# Estimating Jaccard Index with Missing Observations: A Matrix Calibration Approach

**Wenye Li**
Macao Polytechnic Institute
Macao SAR, China
wyli@ipm.edu.mo

## Abstract

The Jaccard index is a standard statistics for comparing the pairwise similarity between data samples. This paper investigates the problem of estimating a Jaccard index matrix when there are missing observations in data samples. Starting from a Jaccard index matrix approximated from the incomplete data, our method calibrates the matrix to meet the requirement of positive semi-definiteness and other constraints, through a simple alternating projection algorithm. Compared with conventional approaches that estimate the similarity matrix based on the imputed data, our method has a strong advantage in that the calibrated matrix is guaranteed to be closer to the unknown ground truth in the Frobenius norm than the un-calibrated matrix (except in special cases they are identical). We carried out a series of empirical experiments and the results confirmed our theoretical justification. The evaluation also reported significantly improved results in real learning tasks on benchmark datasets.

## 1 Introduction

A critical task in data analysis is to determine how similar two data samples are. The applications arise in many science and engineering disciplines. For example, in statistical and computing sciences, similarity analysis lays a foundation for cluster analysis, pattern classification, image analysis and recommender systems [15, 8, 17].

A variety of similarity models have been established for different types of data. When data samples can be represented as algebraic vectors, popular choices include cosine similarity model, linear kernel model, and so on [24, 25]. When each vector element takes a value of zero or one, the Jaccard index model is routinely applied, which measures the similarity by the ratio of the number of unique elements common to two samples against the total number of unique elements in either of them [14, 23].

Despite the wide applications, the Jaccard index model faces a non-trivial challenge when data samples are not fully observed. As a treatment, imputation approaches may be applied, which replace the missing observations with substituted values and then calculate the Jaccard index based on the imputed data. Unfortunately, with a large portion of missing observations, imputing data samples often becomes un-reliable or even infeasible, as evidenced in our evaluation.

Instead of trying to fill in the missing values, this paper investigates a completely different approach based on matrix calibration. Starting from an approximate Jaccard index matrix that is estimated from incomplete samples, the proposed method calibrates the matrix to meet the requirement of positive semi-definiteness and other constraints. The calibration procedure is carried out with a simple yet flexible alternating projection algorithm.

The proposed method has a strong theoretical advantage. The calibrated matrix is guaranteed to be better than, or at least identical to (in special cases), the un-calibrated matrix in terms of a shorter Frobenius distance to the true Jaccard index matrix, which was verified empirically as well. Besides, our evaluation of the method also reported improved results in learning applications, and the improvement was especially significant with a high portion of missing values.

A note on notation. Throughout the discussion, a data sample, $A_i$ ($1 \leq i \leq n$), is treated as a set of features. Let $F = \{f_1, \cdots, f_d\}$ be the set of all possible features. Without causing ambiguity, $A_i$ also represents a binary-valued vector. If the $j$-th ($1 \leq j \leq d$) element of vector $A_i$ is one, it means $f_j \in A_i$ (feature $f_j$ belongs to sample $A_i$); if the element is zero, $f_j \notin A_i$; if the element is marked as *missing*, it remains unknown whether feature $f_j$ belongs to sample $A_i$ or not.

## 2 Background

### 2.1 The Jaccard index

The Jaccard index is a commonly used statistical indicator for measuring the pairwise similarity [14, 23]. For two nonempty and finite sets $A_i$ and $A_j$, it is defined to be the ratio of the number of elements in their intersection against the number of elements in their union:

$$J_{ij}^* = \frac{|A_i \cap A_j|}{|A_i \cup A_j|}$$

where $|\cdot|$ denotes the cardinality of a set.

The Jaccard index has a value of $0$ when the two sets have no elements in common, $1$ when they have exactly the same elements, and strictly between $0$ and $1$ otherwise. The two sets are more similar (have more common elements) when the value gets closer to $1$.

For $n$ sets $A_1, \cdots, A_n$ ($n \geq 2$), the Jaccard index matrix is defined as an $n \times n$ matrix $J^* = \left\{J_{ij}^*\right\}_{i,j=1}^n$. The matrix is symmetric and all diagonal elements of the matrix are $1$.

### 2.2 Handling missing observations

When data samples are fully observed, the accurate Jaccard index can be obtained trivially by enumerating the intersection and the union between each pair of samples if both the number of samples and the number of features are small. For samples with a large number of features, the index can often be approximated by MinHash and related methods [5, 18], which avoid the explicit counting of the intersection and the union of the two sets.

When data samples are not fully observed, however, obtaining the accurate Jaccard index generally becomes infeasible. One naïve approximation is to ignore the features with missing values. Only those features that have no missing values in all samples are used to calculate the Jaccard index. Obviously, for a large dataset with missing-at-random features, it is very likely that this method will throw away all features and therefore does not work at all.

The mainstream work tries to replace the missing observations with substituted values, and then calculates the Jaccard index based on the imputed data. Several simple approaches, including **zero**, **median** and $k$-nearest neighbors (**kNN**) methods, are popularly used. A missing element is set to zero, often implying the corresponding feature does not exist in a sample. It can also be set to the median value (or the mean value) of the feature over all samples, or sometimes over a number of nearest neighboring instances.

A more systematical imputation framework is based on the classical expectation maximization (**EM**) algorithm [6], which generalizes maximum likelihood estimation to the case of incomplete data. Assuming the existence of un-observed latent variables, the algorithm alternates between the expectation step and the maximization step, and finds maximum likelihood or maximum a posterior estimates of the un-observed variables. In practice, the imputation is often carried out through iterating between learning a mixture of clusters of the filled data and re-filling missing values using cluster means, weighted by the posterior probability that a cluster generates the samples [11].

# 3 Solution

Our work investigates the Jaccard index matrix estimation problem for incomplete data. Instead of throwing away the un-observed features or imputing the missing values, a completely different solution based on matrix calibration is designed.

## 3.1 Initial approximation

For a sample $A_i$, denote by $O_i^+$ the set of features that are known to be in $A_i$, and denote by $O_i^-$ the set of features that are known to be not in $A_i$. Let $O_i = O_i^+ \cup O_i^-$. If $O_i = F$, $A_i$ is *fully observed* without missing values; otherwise, $A_i$ is *not fully observed* with missing values. The complement of $O_i$ with respect to $F$, denoted by $\overline{O_i}$, gives $A_i$'s unknown features and missing values.

For two samples $A_i$ and $A_j$ with missing values, we approximate their Jaccard index by:

$$J_{ij}^0 = \frac{\left|\left(O_i^+ \cap O_j\right) \cap \left(O_j^+ \cap O_i\right)\right|}{\left|\left(O_i^+ \cap O_j\right) \cup \left(O_j^+ \cap O_i\right)\right|} = \frac{\left|O_i^+ \cap O_j^+\right|}{\left|\left(O_i^+ \cap O_j\right) \cup \left(O_j^+ \cap O_i\right)\right|}$$

Here we assume that each sample has at least one observed feature. It is obvious that $J_{ij}^0$ is equal to the ground truth $J_{ij}^*$ if the samples are fully observed.

There exists an interval $[\ell_{ij}, \mu_{ij}]$ that the true value $J_{ij}^*$ lies in, where

$$\ell_{ij} = \begin{cases} 1, & \text{if } i = j \\ \dfrac{\left|O_i^+ \cap O_j^+\right|}{\left|\overline{O_i^- \cap O_j^-}\right|}, & \text{otherwise} \end{cases}$$

and

$$\mu_{ij} = \begin{cases} 1, & \text{if } i = j \\ \dfrac{\left|\overline{O_i^- \cup O_j^-}\right|}{\left|\overline{O_i \cup O_j} \cup O_i^+ \cup O_j^+\right|}, & \text{otherwise} \end{cases}.$$

The lower bound $\ell_{ij}$ is obtained from the extreme case of setting the missing values in a way that the two sets have the fewest features in their intersection while having the most features in their union. On the contrary, the upper bound $\mu_{ij}$ is obtained from the other extreme. When the samples are fully observed, the interval shrinks to a single point $\ell_{ij} = \mu_{ij} = J_{ij}^*$.

## 3.2 Matrix calibration

Denote by $J^* = \left\{J_{ij}^*\right\}_{ij=1}^n$ the true Jaccard index matrix for a set of data samples $\{A_1, \cdots, A_n\}$, we have [2]:

**Theorem 1.** *For a given set of data samples, its Jaccard index matrix $J^*$ is positive semi-definite.*

For data samples with missing values, the matrix $J^0 = \left\{J_{ij}^0\right\}_{ij=1}^n$ often loses positive semi-definiteness. Nevertheless, it can be calibrated to ensure the property by seeking an $n \times n$ matrix $J = \{J_{ij}\}_{ij=1}^n$ to minimize:

$$L_0 (J) = \left\|J - J^0\right\|_F^2$$

subject to the constraints:

$$J \succeq 0, \text{ and, } \ell_{ij} \leq J_{ij} \leq \mu_{ij} \ (1 \leq i, j \leq n)$$

where $J \succeq 0$ requires $J$ to be positive semi-definite and $\|\cdot\|_F$ denotes the Frobenius norm of a matrix and $\|J\|_F^2 = \sum_{ij} J_{ij}^2$.

Let $M_n$ be the set of $n \times n$ symmetric matrices. The feasible region defined by the constraints, denoted by $R$, is a nonempty closed and convex subset of $M_n$. Following standard results in optimization theory [20, 3, 10], the problem of minimizing $L_0 (J)$ is convex. Denote by $P_R$ the projection onto $R$. Its unique solution is given by the projection of $J_0$ onto $R$: $J_R^0 = P_R \left(J^0\right)$.

For $J_R^0$, we have:

**Theorem 2.** $\left\|J^* - J_R^0\right\|_F^2 \leq \left\|J^* - J^0\right\|_F^2$. *The equality holds iff $J^0 \in R$, i.e., $J^0 = J_R^0$.*

*Proof.* Define an inner product on $M_n$ that induces the Frobenius norm:

$$\langle X, Y \rangle = trace\left(X^T Y\right), \text{ for } X, Y \in M_n.$$

Then

$$
\begin{aligned}
& \left\|J^* - J^0\right\|_F^2 \\
=\ & \left\|\left(J^* - J_R^0\right) - \left(J^0 - J_R^0\right)\right\|_F^2 \\
=\ & \left\|J^* - J_R^0\right\|_F^2 + \left\|J^0 - J_R^0\right\|_F^2 - 2\left\langle J^* - J_R^0, J^0 - J_R^0\right\rangle \\
\geq\ & \left\|J^* - J_R^0\right\|_F^2 - 2\left\langle J^* - J_R^0, J^0 - J_R^0\right\rangle \\
\geq\ & \left\|J^* - J_R^0\right\|_F^2
\end{aligned}
$$

The second "$\geq$" holds due to the Kolmogrov's criterion, which states that the projection of $J^0$ onto $R$, $J_R^0$, is unique and characterized by:

$$J_R^0 \in R, \text{ and } \left\langle J - J_R^0, J^0 - J_R^0\right\rangle \leq 0 \text{ for all } J \in R.$$

The equality holds iff $\left\|J^0 - J_R^0\right\|_F^2 = 0$ and $\left\langle J^* - J_R^0, J^0 - J_R^0\right\rangle = 0$, i.e., $J^0 = J_R^0$. $\qquad\square$

This key observation shows that projecting $J^0$ onto the feasible region $R$ will produce an improved estimate towards $J^*$, although this ground truth matrix remains unknown to us.

### 3.3 Projection onto subsets

Based on the results in Section 3.2, we are to seek a minimizer to $L_0\left(J\right)$ to improve the estimate $J^0$. Define two nonempty closed and convex subsets of $M_n$:

$$S = \left\{X | X \in M_n, X \succeq 0\right\}$$

and

$$T = \left\{X | X \in M_n, \ell_{ij} \leq X_{ij} \leq \mu_{ij} \left(1 \leq i, j \leq n\right)\right\}.$$

Obviously $R = S \cap T$. Now our minimization problem becomes finding the projection of $J^0$ onto the intersection of two sets $S$ and $T$ with respect to the Frobenius norm. This can be done by studying the projection onto the two sets individually. Denote by $P_S$ the projection onto $S$, and $P_T$ the projection onto $T$. For projection onto $T$, a straightforward result based on the Kolmogrov's criterion is:

**Theorem 3.** *For a given matrix $X \in M_n$, its projection onto $T$, $X_T = P_T\left(X\right)$, is given by*

$$
\left(X_T\right)_{ij} = \begin{cases} X_{ij}, & \text{if } \ell_{ij} \leq X_{ij} \leq \mu_{ij} \\ \ell_{ij}, & \text{if } X_{ij} < \ell_{ij} \\ \mu_{ij}, & \text{if } X_{ij} > \mu_{ij} \end{cases}.
$$

For projection onto $S$, a well known result is the following [12, 16, 13]:

**Theorem 4.** *For $X \in M_n$ and its singular value decomposition $X = U\Sigma V^T$ where $\Sigma = diag\left(\lambda_1, \cdots, \lambda_n\right)$, the projection of $X$ onto $S$ is given by: $X_S = P_S\left(X\right) = U\Sigma' V^T$ where $\Sigma' = diag\left(\lambda_1', \cdots, \lambda_n'\right)$ and*

$$
\lambda_i' = \begin{cases} \lambda_i, & \text{if } \lambda_i \geq 0 \\ 0, & \text{otherwise} \end{cases}.
$$

The matrix $X_S = P_S\left(X\right)$ gives the positive semi-definite matrix that most closely approximates $X$ with respect to the Frobenius norm.

### 3.4 Dykstra's algorithm

To study the orthogonal projection onto the intersection of subspaces, a classical result is von Neumann's alternating projection algorithm. Let $H$ be a Hilbert space with two closed subspaces $C_1$ and $C_2$. The orthogonal projection onto the intersection $C_1 \cap C_2$ can be obtained by the product of the two projections $P_{C_1} P_{C_2}$ when the two projections commute ($P_{C_1} P_{C_2} = P_{C_2} P_{C_1}$). When they do not commute, the work shows that for each $x^0 \in H$, the projection of $x^0$ onto the intersection can be obtained by the limit point of a sequence of projections onto each subspace respectively: $\lim_{k \to \infty} (P_{C_2} P_{C_1})^k (x^0) = P_{C_1 \cap C_2} (x^0)$. The algorithm generalizes to any finite number of subspaces and projections onto them.

Unfortunately, different from the application in [19], in our problem both $S$ and $T$ are not subspaces but subsets, and von Neumann's convergence result does not apply. The limit point of the generated sequence may converge to non-optimal points.

To handle the difficulty, Dykstra extended von Neumann's work and proposed an algorithm that works with subsets [9]. Consider the case of $C = \bigcap_{i=1}^{r} C_i$ where $C$ is nonempty and each $C_i$ is a closed and convex subset in $H$. Assume that for any $x \in H$, obtaining $P_C(x)$ is hard, while obtaining each $P_{C_i}(x)$ is easy. Starting from $x^0 \in H$, Dykstra's algorithm produces two sequences, the iterates $\{x_i^k\}$ and the increments $\{I_i^k\}$. The two sequences are generated by:

$$
\begin{aligned}
x_0^k &= x_r^{k-1} \\
x_i^k &= P_{C_i}\left(x_{i-1}^k - I_i^{k-1}\right) \\
I_i^k &= x_i^k - \left(x_{i-1}^k - I_i^{k-1}\right)
\end{aligned}
$$

where $i = 1, \cdots, r$ and $k = 1, 2, \cdots$. The initial values are given by $x_r^0 = x^0, I_i^0 = 0$.

The sequence of $\{x_i^k\}$ converges to the optimal solution with a theoretical guarantee [9, 10].

**Theorem 5.** *Let $C_1, \cdots, C_r$ be closed and convex subsets of a Hilbert space $H$ such that $C = \bigcap_{k=1}^{r} C_k \neq \Phi$. For any $i = 1, \cdots, r$ and any $x^0 \in H$, the sequence $\{x_i^k\}$ converges strongly to $x_C^0 = P_C(x^0)$ (i.e. $\left\| x_i^k - x_C^0 \right\| \to 0$ as $k \to \infty$).*

The convergent rate of Dykstra's algorithm for polyhedral sets is linear [7], which coincides with the convergence rate of von Neumann's alternating projection method.

### 3.5 An iterative method

Based on the discussion in Section 3.4, we have a simple approach, shown in Algorithm 1, that finds the projection of an initial matrix $J^0$ onto the nonempty set $R = S \cap T$. Here the projections onto $S$ and $T$ are given by the two theorems in Section 3.3. The algorithm stops when $J^k$ falls into the feasible region or when a maximal number of iterations is achieved. For practical implementation, a more robust stopping criterion can be adopted [1].

### 3.6 Related work

It is a known study in mathematical optimization field to find a positive semi-definite matrix that is closest to a given matrix. A number of methods have been proposed recently. The idea of alternating projection method was firstly applied in a financial application [13]. The problem can also be phrased as a semi-definite programming (SDP) model [13] and be solved via the interior-point method. In the work of [21] and [4], the quasi-Newton method and the projected gradient method to the Lagrangian dual of the original problem were applied, which reported faster results than the SDP formulation. An even faster Newton's method was developed in [22] by investigating the dual problem, which is unconstrained with a twice continuously differentiable objective function and has a quadratically convergent solution.

---

**Algorithm 1** Projection onto $R = S \cap T$

---

**Require:** Initial matrix $J^0$
   $k = 0$
   $J_T^0 = J^0$
   $I_S^0 = 0$
   $I_T^0 = 0$
   **while** NOT CONVERGENT **do**
      $J_S^{k+1} = P_S \left( J_T^k - I_S^k \right)$
      $I_S^{k+1} = J_S^{k+1} - \left( J_T^k - I_S^k \right)$
      $J_T^{k+1} = P_T \left( J_S^{k+1} - I_T^k \right)$
      $I_T^{k+1} = J_T^{k+1} - \left( J_S^{k+1} - I_T^k \right)$
      $k = k + 1$
   **end while**
   **return** $J^k = J_T^k$

---

## 4    Evaluation

To evaluate the performance of the proposed method, four benchmark datasets were used in our experiments.

- *MNIST*: a grayscale image database of handwritten digits ("0" to "9"). After binarization, each image is represented as a $784$-dimensional 0-1 vector.

- *USPS*: another grayscale image database of handwritten digits. After binarization, each image is represented as a $256$-dimensional 0-1 vector.

- *PROTEIN*: a bioinformatics database with three classes of instances. Each instance is represented as a sparse $357$-dimensional 0-1 vector.

- *WEBSPAM*: a dataset with both spam and non-spam web pages. Each page is represented as a 0-1 vector. The data are highly sparse. On average one vector has about $4,000$ non-zero values out of more than 16 million features.

Our experiments have two objectives. One is to verify the effectiveness of the proposed method in estimating the Jaccard index matrix by measuring the derivation of the calibrated matrix from the ground truth in Frobenius norm. The other is to evaluate the performance of the calibrated matrix in general learning applications. The comparison is made against the popular imputation approaches listed in Section 2.2, including the **zero**, **kNN** and **EM** [1] approaches. (As the **median** approach gave very similar performance as the **zero** approach, its results were not reported separately.)

### 4.1    Jaccard index matrix estimation

The experiment was carried out under various settings. For each dataset, we experimented with $1,000$ and $10,000$ samples respectively. For each sample, different portions (from $10\%$ to $90\%$) of feature values were marked as missing, which was assumed to be "missing at random" and all features had the same probability of being marked.

As mentioned in Section 3, for the proposed calibration approach, an initial Jaccard index matrix was firstly built based on the incomplete data. Then the matrix was calibrated to meet the positive semi-definite requirement and the lower and upper bounds requirement. While for the imputation approaches, the Jaccard index matrix was calculated directly from the imputed data.

Note that for the **kNN** approach, we iterated different $k$ from 1 to 5 and the best result was collected, which actually overestimated its performance. Under some settings, the results of the **EM** approach were not available due to its prohibitive computational requirement to our platform.

The results are presented through the comparison of mean square deviations from the ground truth of the Jaccard index matrix $J^*$. For an $n \times n$ estimated matrix $J'$, its mean square deviation from

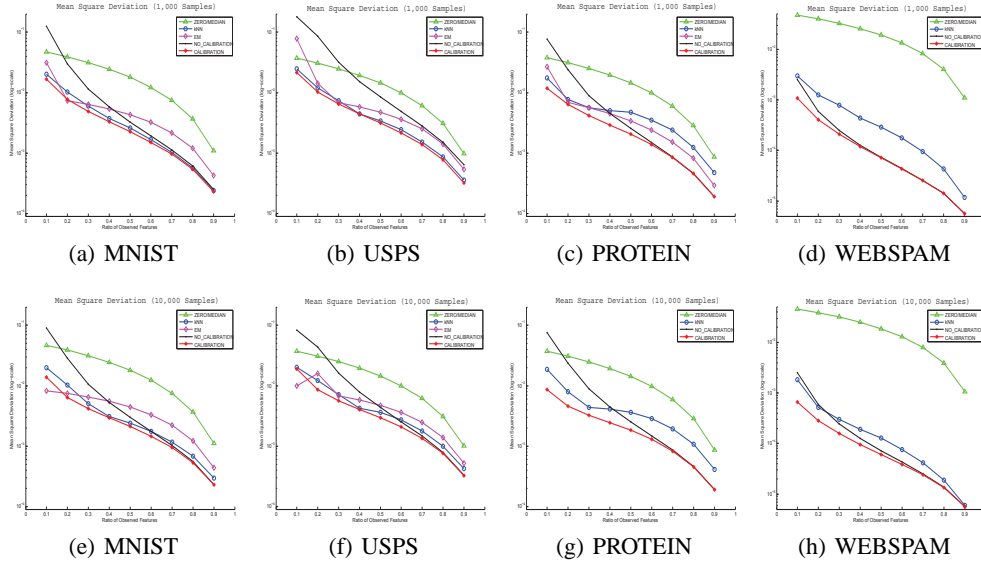

Figure 1: Mean square deviations from the ground truth on benchmark datasets by different methods. Horizontal: percentages of observed values (from 10% to 90%); Vertical: mean square deviations in log-scale. (a)-(d): $1,000$ samples; (e)-(f): $10,000$ samples. (For better visualization effect of the results shown in color, the reader is referred to the soft copy of this paper.)

$J^*$ is defined as the square Frobenius distance between the two matrices, divided by the number of elements, i.e., $\frac{\sum_{ij=1}^{n}\left(J'_{ij}-J^*_{ij}\right)^2}{n^2}$. In addition to the comparison with the popular approaches, the mean square deviation between the un-calibrated matrix $J^0$ and $J^*$, shown as **NO_CALIBRATION**, is also reported as a baseline.

Figure 1 shows the results. It can be seen that the calibrated matrices reported the smallest derivation from the ground truth in nearly all experiments. The improvement is especially significant when the ratio of observed features is low (the missing ratio is high). It is guaranteed to be no worse than the un-calibrated matrix. As evidenced in the results, for all the imputation approaches, there is no such a guarantee.

## 4.2 Supervised learning

Knowing the improved results in reducing the deviation from the ground truth matrix, we would like to further investigate whether this improvement indeed benefits practical applications, specifically in supervised learning.

We applied the calibrated results in nearest neighbor classification tasks. Given a training set of labeled samples, we tried to predict the labels of the samples in the testing set. For each testing sample, its label was determined by the label of the sample in the training set that had the largest Jaccard index value with it.

Similarly the experiment was carried out with $1,000/10,000$ samples and different portions of missing values from $10\%$ to $90\%$ respectively. In each run, $90\%$ of the samples were randomly chosen as the training set and the remaining $10\%$ were used as the testing set. The mean and standard deviation of the classification errors in $1,000$ runs were reported. As a reference, the results from the ground truth matrix $J^*$, shown as **FULLY_OBSERVED**, were also included.

Figure 2 shows the results. Again the matrix calibration method reported evidently improved results over the imputation approaches in most experiments. The improvement verified the benefits brought by the reduced deviation from the true Jaccard index matrix, and therefore justified the usefulness of the proposed method in learning applications.

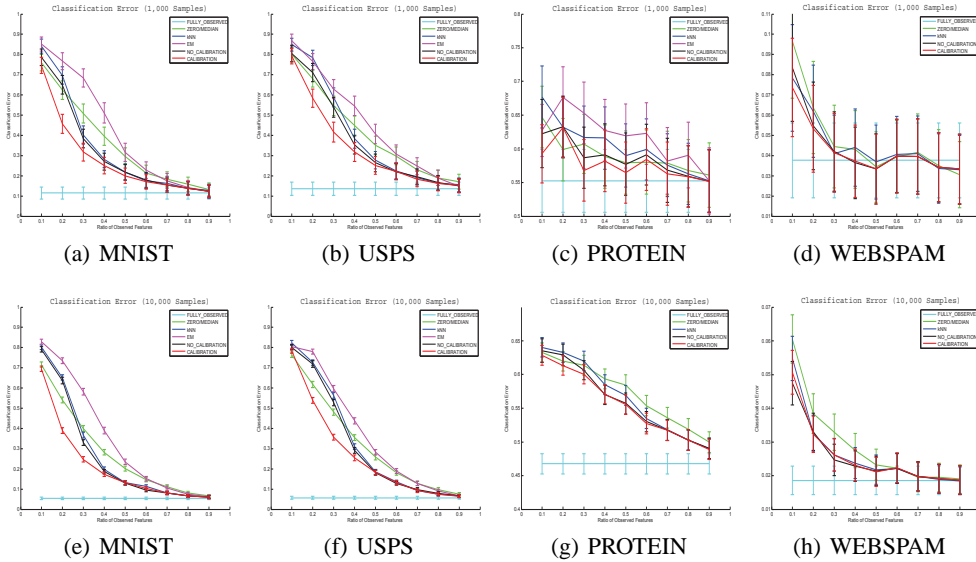

|  |  |  |  |
|---|---|---|---|
| (a) MNIST | (b) USPS | (c) PROTEIN | (d) WEBSPAM |
| (e) MNIST | (f) USPS | (g) PROTEIN | (h) WEBSPAM |

Figure 2: Classification errors on benchmark datasets by different methods. Horizontal: percentage of observed values (from 10% to 90%); Vertical: classification errors. (a)-(d): $1,000$ samples; (e)-(f): $10,000$ samples. (For better visualization effect of the results shown in color, the reader is referred to the soft copy of this paper.)

## 5   Discussion and conclusion

The Jaccard index measures the pairwise similarity between data samples, which is routinely used in real applications. Unfortunately in practice, it is non-trivial to estimate the Jaccard index matrix for incomplete data samples. This paper investigates the problem, and proposes a matrix calibration approach in a way that is completely different from the existing methods. Instead of throwing away the unknown features or imputing the missing values, the proposed approach calibrates any approximate Jaccard index matrix by ensuring the positive semi-definite requirement on the matrix. It is theoretically shown and empirically verified that the approach indeed brings about improvement in practical problems.

One point that is not particularly addressed in this paper is the computational complexity issue. We adopted a simple alternating projection procedure based on Dykstra's algorithm. The computational complexity of the algorithm heavily depends on the successive matrix decompositions. It is expensive when the size of the matrix becomes large. Calibrating a Jaccard index matrix for $1,000$ samples can be finished in seconds of time on our platform, while calibrating a matrix for $10,000$ samples quickly increases to more than an hour. Further investigations for faster solutions are thus necessary for scalability.

Actually, there is a simple divide-and-conquer heuristic to calibrate a large matrix. Firstly divide the matrix into small sub-matrices. Then calibrate each sub-matrix to meet the constraints. Finally merge the results. Although the heuristic may not give the optimal result, it also guarantees to produce a matrix better than or identical to the un-calibrated matrix. The heuristic runs with high parallel efficiency and easily scales to very large matrices. The detailed discussion is omitted here due to the space limit.

#### Acknowledgments

The work is supported by The Science and Technology Development Fund (Project No. 006/2014/A), Macao SAR, China.

## Footnotes

[1]ftp://ftp.cs.toronto.edu/pub/zoubin/old/EMcode.tar.Z

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
