[Reviews · NeurIPS 2015]

Submitted by Assigned_Reviewer_1

The paper was clear and concise. The idea is original as far as the reviewer is aware, and after a modest amount of literature search. The experimental results are convincing enough to warrant considering the approach for small to medium sized problems, with sample sizes in the order of 1K-10K, and the ratio of observed features is in the .2-.5 range. It's not clear from the experimental evaluation, however, whether the approach is as effective, compared to the other imputation approaches, on larger problems (larger similarity matrices) when projecting onto the set of positive semi-definite matrices - which relies on computation of SVD - becomes intractable or requires approximate methods (the authors do not clarify whether approximate SVD was used in the experiments). As the authors point out, the matrix decomposition is a major computational bottleneck and limitation of this method, and as such the reviewer would argue that this would make it an unappealing method to apply to many real world problems where sample sizes are much larger that 10K. This (what happens with larger data sets) is a major concern.

Another concern is the relevance of this whole apparatus. I am not aware, off the cuff, of a practical situation in which one would want to compute Jaccard similarity matrix while facing missing values in the samples. It is quite possible that such a situation comes up, or even is very important, but the authors fail to provide any practical examples. All of the experiments are on data sets (like MNIST or USPS) in which the values are dropped artificially. Obviously, such experiments are crucial to evaluate the accuracy of the proposed method under controlled circumstances, and to measure its sensitivity to various parameters. But equally crucial, I think, is to provide at least one example in which real data, with real underlying phenomenon of interest (here the missing values) is examined. Since in real data missing values are most likely not distributed uniformly (see below more on this) it would be potentially a different "game" and the outcome, in particular the relative standing of the proposed method vs. imputation, is uncertain. I'd welcome report on any additional experiment along these lines that the authors can include in the rebuttal.

Another suggestion for improvement, related to the comment above: consider cases other than uniformly random missing features. At some point in continuum from high to low entropy distributions one could expect imputation via kNN to outperform.

Summary: The authors present a novel approach to estimating the Jaccard index matrix - a pairwise sample similarity matrix - in the presence of incomplete samples. This approach is novel and the experimental evaluation suggests it may be useful for real world situations where one needs to compute Jaccard indices on examples with missing values. However, it is not clear if and when this situation actually arises; furthermore, there is a concern about feasibility of the proposed approach on large data sets.

Submitted by Assigned_Reviewer_2

This paper addresses the problem of estimating the Jaccard index similarity measure between two data samples in the face of missing observations.

The approach is to estimate a complete Jaccard matrix while respecting the constraint of positive semi-definiteness.

This problem is convex and has a known solution.

The method is guaranteed to return a Jaccard matrix closer to the true matrix in terms of Frobenius norm.

The authors demonstrate that their method leads to improved performance on standard datasets with missing data relative to commonly-used heuristics.

Quality:

Well-executed study Clarity:

Paper is very clear. Spacing in equations on Page 3 should be adjusted so that the \bar symbols do not merge with the fraction bar. Originality:

Not my area, but it seems like this important problem has not been addressed previously. Significance:

Not earth-shattering, but an important, practical contribution.
Summary: Competent study on technical issue dealing with missing data.

Practical results.

Submitted by Assigned_Reviewer_3

Given two sets of binary feature measurements, the Jaccard similarity index measures the ratio of intersection over the union of the these two sets. This paper deals with the problem of computing the Jaccard index when the value of a subset of features is unknown.

The idea is to operate on the Jaccard matrix, i.e., a matrix that combines Jaccard indices between sets of observations. The authors exploit the fact that this matrix must be positive semi-definite to propose a method that deals with missing features by calibrating the Jaccard matrix. This is done by finding its nearest positive semi-definite matrix. They proposed to use an optimisation method based on Dykstra's algorithm. The paper shows that the matrices obtained with this method are guaranteed to be no worse than the un-calibrated matrices. This is shown both theoretically and in practice, by comparing the obtained matrices against ground truth matrices and in a classification set up.

I'm happy with the clarity and overall quality of the paper, however I am not so sure about its originality. Although the idea of using a matrix calibration approach is novel, the method proposed is based on existing algorithms. Furthermore, section 3.6 lists some alternative methods, but it does not clearly state what are the advantages of the method proposed in relation to those methods. It would be interesting to see how one of those methods would perform in the experimental benchmarks.

The significance of this paper is quite high, as it has the problem approached is very common and most researchers tend to use naive approaches do deal with missing measurements.
Summary: This paper is clear and has a very high significance, but I am not convinced if the experimental validation is broad enough as it does not present a comparison against other calibration methods.

Submitted by Assigned_Reviewer_4

Summary: The paper proposes a matrix calibration approach for estimating Jaccard index with missing observations. The calibration approach is based on the property of the positive definiteness of the Jaccard index. The propose algorithm is very clean(free of parameter tuning) and convergence is guaranteed. However, just as the author point out, the algorithm is computationally expensive which limits its potential applications.

Originality and significance: The novelty of this paper is the discovery that the property of positive definiteness of the Jaccard index

can be exploited in the context of missing observation which leads to a well-studied optimization problem. The success of the method relies on a good initial guess($J^0$). The lack of theoretical justification of the initial guess $J^0$ makes the approach somewhat heuristic. Also, the paper would be more insightful and more significant if the author could establish the validity of the method under some assumption about the probability distribution of the data.

Clarity: The paper is well-written.
Summary: This paper proposes a new approach for estimating Jaccard Index with missing data. The paper provides new insights and is well-written.

Submitted by Assigned_Reviewer_5

This paper proposes a method to estimate Jaccard index when it has missing values. The proposed method is a convex problem, and optimal result can be found. Empirical tests showed that the proposed calibration works in practical datasets. - It is nice approach to estimate similarity matrix of Jaccard index Instead of solving the problem to fill out the missing values. - The evaluation is not enough to show whether the proposed estimation works well or not because only nearest neighbor classification tasks were tested.
Summary: While a method to treat missing values is an important problem, the optimal method depends on case. Need to describe clear purposes to estimate the missing values.

Author Feedback
Author rebuttal: Thanks for all reviewers' constructive comments in improving the paper.

Relating to the concerns in the review, the following points are further clarified. (A supplemental file with details is available at: https://drive.google.com/file/d/0B1Oy7KyELrDVVVQzWmZVeFl4X2s/view?usp=sharing)

1. Scalability:
Compared with imputation approaches, our method does not suffer from "the curse of dimensionality" (at least not directly). On problems with large dimensions (e.g. WEBSPAM with 16M dimensions), it is much faster than some imputation approaches.
On the other hand, as mentioned in the paper, our method relies on successive matrix SVDs. It is fairly quick for small problems (in seconds/within a minute for 1K data), but non-trivial for large problems (over an hour for 10K data). For even larger problems, direct applying the method can be infeasible. (Thanks for suggesting approximate SVD to us. We'll evaluate its potential application in coming work.)

To handle the bottleneck, a divide-and-conquer heuristic works. Firstly divide a large matrix into small sub-matrices. Then calibrate each sub-matrix. Finally merge the results. Although the heuristic may not produce the optimal result, it also guarantees to improve the un-calibrated matrix (Ref. supplemental file for details).
The heuristic executes with high parallel efficiency and scales to large problems. As a test, we applied the heuristic to the full MNIST dataset (70K data), and finished the experiment within 20 hours on a workstation with 12 CPU cores.

2. Application/Experiment/Comparison:
Jaccard index (J-index) is commonly used to measure the similarity of data items (http://infolab.stanford.edu/~ullman/mmds/ch3.pdf). We believe calibrating an approximate J-index matrix has a variety of applications. Our paper chose kNN classification on incomplete data as an example and reported improved results.

Another direct application is in Recommendation Systems (RS, http://infolab.stanford.edu/~ullman/mmds/ch9.pdf). An RS has two entity classes: users and items. Each user has preferences for certain items. The data is represented as a utility matrix. Each entry, if known, gives for each user-item pair a value that represents the rating of the user for the item. The matrix is sparse; most entries are unknown. The goal of RS is to predict these unknowns.
Two common methods in RS include item-based collaborative filtering (CF) and user-based CF. The two CFs are based on item-item and user-user similarity matrices, where J-index is often adopted. As suggested in the review, we further tested our method in an RS application with "real" missing values. We found, after calibrating J-index matrices, the CFs had consistently improved results (Ref. supplemental file).

It is noteworthy that, with slight modification, the calibration also applies to other models such as cosine similarity and linear kernel. We expect further work along the line.

Comparison with related optimization methods: Recall in our problem, there are generally n equality constraints and n(n-1)/2 inequality/box constraints. The large number of box constraints bring much trouble to related methods. Specifically, we were unable to finish running the SDP model [13] on 10K data; [19] and [20] do not support the box constraints directly; the method of [4] encountered slower convergence in our problems (mainly because, as shown in [4], the method needs to set a very small step size to guarantee the convergence in the presence of many box constraints).

3. On prior knowledge:
We totally agree that in many applications with proper domain-specific knowledge, imputation approaches such as kNN are very successful.

As suggested in the review and thanks for the suggestion, prior knowledge/assumption (such as the probability distribution of the data) can be exploited to extend and justify our work. Specifically our supplemental file shows such a trial.

4. On kNN classification:
Sorry for any confusion here.
Simply for classification purposes, the affinity matrix needs not to be positive definite.
While in our experiment, the data is not complete. Thus the J-index affinity matrix is only an approximation and not necessarily p.d. By calibrating the approximate J-index matrix to be p.d., it gets closer to the true J-index matrix. We hope to verify if this calibrated matrix helps in improving the classification accuracy.